# Airborne Platform Three-Dimensional Positioning Method Based on Interferometric Synthetic Aperture Radar Interferogram Matching

Lanyu Li [1,2], Yachao Wang [1,*], Bingnan Wang [1] and Maosheng Xiang [1]

1   National Key Laboratory of Microwave Imaging Technology, Aerospace Information Research Institute, Chinese Academy of Sciences, Beijing 100190, China; lilanyu21@mails.ucas.ac.cn (L.L.)
2   School of Electronic, Electrical and Communication Engineering, University of Chinese Academy of Sciences, Beijing 100049, China
*   Correspondence: wangyc@aircas.ac.cn

**Abstract:** As the demand for precise navigation of aircraft increases in modern society, researching high-precision, high-autonomy navigation systems is both theoretically valuable and practically significant. Because the inertial navigation system (INS) has systematic and random errors, its output information diverges. Therefore, it is necessary to combine them with other navigation systems for real-time compensation and correction of these errors. The SAR matching positioning and navigation system uses synthetic aperture radar (SAR) image matching for platform positioning and compensates for the drift caused by errors in the inertial measurement unit (IMU). Images obtained by SAR are matched with digital landmark data, and the platform's position is calculated based on the SAR imaging geometry. However, SAR matching positioning faces challenges due to seasonal variations in SAR images, the need for typical landmarks for matching, and the lack of elevation information in two-dimensional SAR image matching. This paper proposes an airborne platform positioning method based on interferometric SAR (InSAR) interferogram matching. InSAR interferograms contain terrain elevation information, are less affected by seasonal changes, and provide higher positioning accuracy and robustness. By matching real-time InSAR-processed interferograms with simulated interferograms using a digital elevation model (DEM), three-dimensional position information about the matching points has been obtained. Subsequently, a three-dimensional positioning model for the platform has bene established using the unit line-of-sight vector decomposition method. In actual flight experiments using an FMCW Ku-band Interferometric SAR system, the proposed platform positioning framework demonstrated its ability to achieve precise positioning in the absence of signals from the global navigation satellite system (GNSS).

**Keywords:** InSAR; interferogram matching; three-dimensional positioning

## 1. Introduction

Radar-based positioning and navigation methods include one-dimensional terrain contour matching [1,2] and two-dimensional SAR image matching navigation [3,4]. Radar, as an additional sensor, is used to address the drift problem caused by errors in the IMU, especially in situations where GNSS signals are lost or interrupted. The principle of digital terrain matching navigation is that as the aircraft flies over certain specific terrain areas along its flight path, it uses radar altimeters to measure the terrain elevation profile along the path. These real-time measurements are then correlated with pre-stored reference maps to determine the aircraft's geographical position. Existing terrain matching methods mainly involve correlating altimeter-measured terrain profiles with those in reference maps, which is a line matching approach that lacks resolution in the cross-track direction, leading to potential mismatches, and requiring areas with typical terrain slope characteristics for matching.

Apart from one-dimensional terrain matching, SAR matching navigation achieves a two-dimensional resolution. It matches real-time radar-acquired images with stored SAR images in its database to correct current positional errors. However, SAR scene matching navigation is affected by seasonal changes in terrain features, requiring typical landmarks for reliable matching performance and is challenging to map in areas with significant terrain variations, leading to poor navigation accuracy in mountainous regions. Researchers have focused on two main areas: one is the matching of heterogeneous SAR images, proposing algorithms tailored to SAR image features to improve matching precision and robustness [5,6]. The other involves using the positional information of SAR matching points and SAR's slant range data to establish the relationship between the aircraft and matching points to calculate the aircraft's position [3,7,8].

InSAR uses interferometry to accurately measure terrain elevation and achieve two-dimensional imaging. It effectively combines the advantages of terrain matching navigation and the SAR image matching navigation methods. InSAR interferometric matching navigation has different working modes. When the terrain undulations in the navigation area are obvious, the InSAR interferometric matching navigation mode is enabled, and the generated interferograms with terrain undulation characteristics are used for matching to achieve platform positioning. When the terrain within the navigation area is relatively flat, single-channel SAR images can be used to complete SAR image matching navigation. Therefore, introducing InSAR technology into positioning and navigation systems offers significant advantages and broad application scenarios. The European Defence Agency (EDA) conducted preliminary analyses of the potential of InSAR-assisted navigation [9–11], exploring InSAR parameters and demonstrating the feasibility of InSAR-assisted navigation. They also studied the impact of position and attitude angle errors on phase and positioning errors, as well as the characteristics of digital terrain models. Subsequently, researchers established a platform position inversion model using matched interferogram offsets and proposed using interferometric phase and Doppler frequency to infer the platform's attitude angles [12,13]. Further, a mountain branch point-based InSAR interferometric fringe matching algorithm was proposed to achieve precise matching between interferometric fringes [14]. Later, researchers began to theoretically analyze high-speed platform positioning based on wrapped InSAR interferograms, providing simulation experiments [15].

While InSAR demonstrates advantages in accuracy and robustness for matching positioning and navigation compared to traditional SAR, there has not been a complete methodological workflow with real flight data validation for InSAR interferogram matching positioning and navigation so far. Most studies have been based on theoretical analysis and simulations. Therefore, this paper proposes a comprehensive three-dimensional platform positioning method based on InSAR interferogram matching, establishing a robust and high-precision platform positioning and navigation system. The rest of the paper is organized as follows: Section 2 presents the proposed three-dimensional positioning framework based on InSAR interferogram matching. Section 3 proposes a method for constructing reference interferograms using external source DEM. Section 4 proposes a feature matching algorithm suitable for interferometric fringe images, achieving precise matching between actual interferograms and reference interferograms. Section 5 proposes a three-dimensional positioning model for the platform based on InSAR altimetry geometry. Section 6 conducts actual InSAR airborne flight experiments, obtaining corresponding positioning results and analyzing positioning accuracy. Finally, Section 7 presents the conclusions of the paper.

## 2. Platform Positioning Framework

InSAR has the ability to generate terrain elevation models and can match these models with DEM. However, considering real-time data processing efficiency, interferograms rather than digital elevation models are used for matching. Figure 1 illustrates a work flow of navigation based on InSAR interferogram matching.

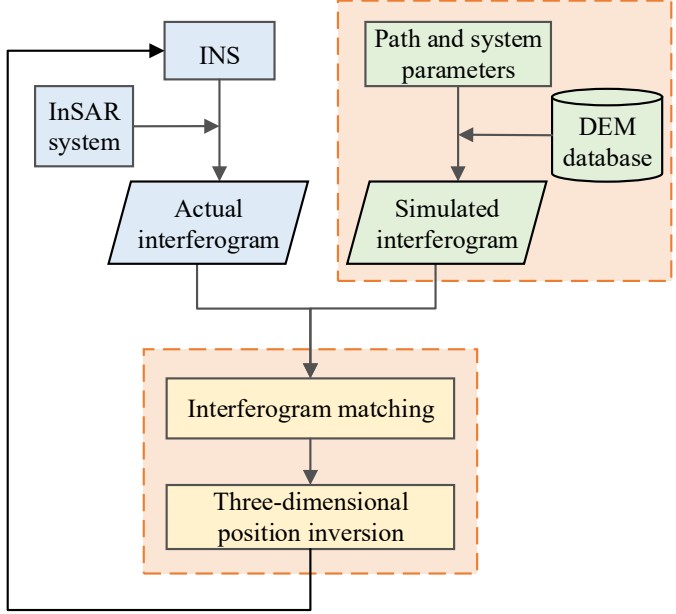

**Figure 1.** InSAR interferogram matching navigation framework. The orange boxes contain the main technical content.

The platform positioning and navigation framework based on InSAR interferogram matching comprises five parts:

(1) First, real-time automatic registration imaging is used to generate actual interferograms, referred to as "actual interferograms".
(2) Subsequently, based on the geometric principles of InSAR imaging, reference digital elevation models are inverted to simulate interferograms, known as "reference interferograms". For detailed information, please refer to Section 3.
(3) Next, a feature matching algorithm is used to match the reference interferograms with the actual interferograms. This process extracts homonymous points between the two interferograms, allowing for the accurate extraction of homonymous point positions from the reference interferograms. For more details, please refer to Section 4.
(4) In the following step, we developed a model to invert and analyze the platform's position. This model utilizes the concept of InSAR three-dimensional inverse positioning. Please see Section 5 for more information.
(5) Finally, the platform's position and the INS positioning information are combined through a filtering process to output the platform's positioning information, achieving long-term autonomous navigation.

## 3. Reference Interference Fringe Generation

The preparation of reference fringe maps is a prerequisite for InSAR matching and positioning, and the accuracy of these reference maps affects both the accuracy of the matching and the positioning. Therefore, higher accuracy in the reference fringe maps can effectively improve positioning accuracy. Currently, there is no interferometric fringe database available, so we use a method of inferring interferometric fringes from DEMs to construct an interferometric fringe library [16,17]. The key lies in calculating the slant range for each point in the DEM of the imaging area based on the geometry of bistatic SAR imaging and the interferometric phase model. Then, the interferogram can be obtained based on the relationship between the difference in slant range and interferometric phase. In actual airborne SAR and navigation guidance applications, it is necessary to consider the impact of the squint angle on the imaging area. We have proposed an algorithm for generating reference interferometric phase maps based on DEM. A flowchart for this process is shown in Figure 2.

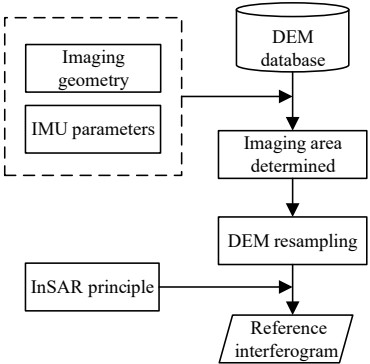

**Figure 2.** Reference interference fringe generation flow chart.

### 3.1. Determination of Imaging Area

The IMU typically records track information in the East-North-Up (ENU) coordinate system, while the determination of the imaging area needs to be carried out in the body coordinate system of the aircraft. Therefore, it is necessary to transform the East-North-Up coordinate system into the aircraft's body coordinate system. The coordinates $(x, y, z)$ represent the body's position in the northeast-up coordinate system. By applying the rotation matrix $\mathbf{R}$, the coordinates in the body coordinate system are obtained as $(x', y', z')$.

$$\begin{bmatrix} x' \\ y' \\ z' \end{bmatrix} = \mathbf{R} \begin{bmatrix} x \\ y \\ z \end{bmatrix} = \begin{bmatrix} \cos\gamma & \sin\gamma & 0 \\ -\sin\gamma & \cos\gamma & 0 \\ 0 & 0 & 1 \end{bmatrix} \begin{bmatrix} x \\ y \\ z \end{bmatrix} \tag{1}$$

After completing the coordinate system transformation, information on the position of the platform can be determined. The next step is to preliminarily determine the imaging area by combining the imaging parameters and the imaging model. The geometric relationship is shown in Figure 3. The projection point of the aircraft on the ground is denoted as shown below. In the InSAR oblique imaging model, there are offsets along the flight direction and perpendicular to the flight direction. We can use the relationship between the oblique angle and the slant range to calculate the offset for each resolution unit:

$$\begin{cases} \Delta x = R \sin\theta_{sq} \\ \Delta y = \sqrt{(R\cos\theta_{sq})^2 - (H - h)^2} \end{cases} \tag{2}$$

where $h$ is the elevation of the ground point. Using the projection point $O$ of the IMU on the ground, as well as the offset along the flight direction $\Delta x$ and the offset perpendicular to the flight direction $\Delta y$, the imaging area in the reference DEM can be calculated.

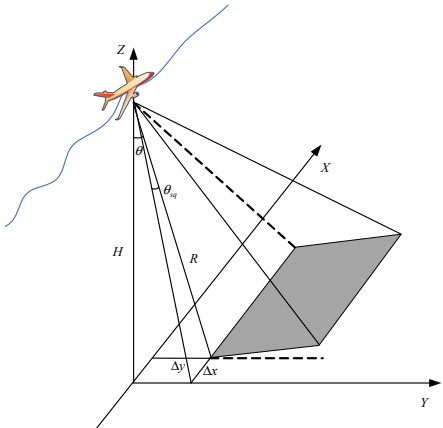

**Figure 3.** Determination of the reference mapping area based on the aircraft imaging geometry.

### 3.2. DEM Resampling

The calculated DEM data are located within the ground range–azimuth plane, while the imaging results of SAR are in the slant range–azimuth plane. Due to terrain undulations, the echo signal from the near-ground point reflecting to the receiver may arrive later than that from the far-ground point. Therefore, it is necessary to resample the DEM based on the size of the slant range, so that the corresponding slant ranges of the DEM are arranged from small to large, as shown in Figure 4:

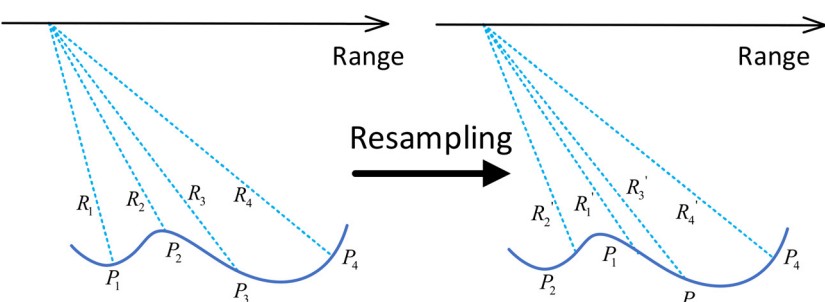

**Figure 4.** DEM slant range resampling schematic, with the left image showing the DEM before resampling and the right image showing the DEM after resampling.

From Figure 4, it can be observed that, due to the undulations in the DEM, the echo from the far point P2 arrives at the receiver earlier than the echo from the near point P1. As a result, the slant range R2 corresponding to the far point is smaller than the slant range R1 corresponding to the near point. After DEM resampling, the DEM in the slant range–azimuth plane is arranged in ascending order based on the slant range. The resampled DEM can be directly used to generate interferometric fringes using the interferometric geometry model. Figure 5 presents the results of DEM resampling, where Figure 5a shows the initial DEM, Figure 5b shows the DEM after slant range resampling, and Figure 5c shows the difference after resampling. It can be observed that, due to the relatively gentle terrain in this area, the DEM difference after resampling is not significant.

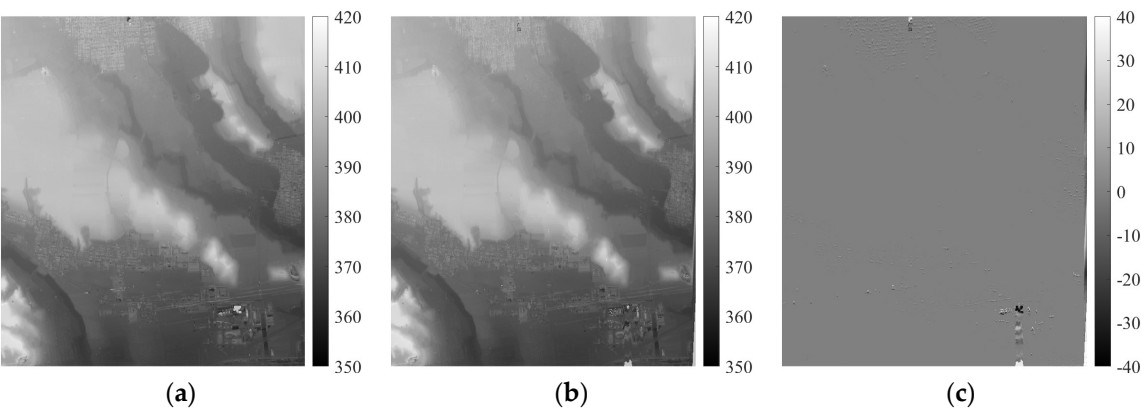

(**a**)           (**b**)           (**c**)

**Figure 5.** Results before and after DEM slope range resampling. (**a**) Original DEM. (**b**) DEM after slant range resampling. (**c**) Difference in DEM after resampling.

### 3.3. Calculate Interferometric Phase and Remove Topographic Contributions

After obtaining the resampled DEM in the imaging area, the interferometric phase of the interferogram can be calculated using the fundamental principles of InSAR to derive the DEM [18]:

$$F = \frac{\Delta R^2}{2RB} - \frac{\Delta R}{B} - \frac{B}{2R} - \sin\left(\alpha - \arccos\frac{H-Z}{R}\right) = 0 \qquad (3)$$

The above formula is a nonlinear equation to calculate the slant range difference $\Delta R$. It can be iteratively solved using optimization methods. After meeting the accuracy conditions, the iterative results are obtained. For an airborne dual-antenna InSAR system, the interferometric phase is directly obtained from the slant range difference:

$$\varphi_a = \frac{2\pi Q \Delta R}{\lambda} \tag{4}$$

Here, $Q$ represents the interferometric mode, where $Q = 1$ corresponds to single-pass interferometry, and $Q = 2$ corresponds to repeat-pass interferometry. The obtained interferometric phase includes the topographic phase, and there is no phase unwrapping at this stage for efficient real-time navigation matching. To avoid excessive line features caused by phase wrapping, we have removed the flat earth phase, and the flat earth phase is:

$$\varphi_g = \frac{2\pi Q}{\lambda}(\sqrt{R^2 + B^2 + 2RB\cos(\alpha - \theta)} - R) \tag{5}$$

Here, $\theta = \arccos(H/R)$, and the symbol $\lfloor \rfloor$ denotes the floor function. The interferometric phase for matching, which includes the wrapped phase, can be expressed as:

$$\varphi = \varphi_a - \varphi_g - 2\pi \left\lfloor \frac{\varphi_a - \varphi_g}{2\pi} \right\rfloor \tag{6}$$

## 4. Interference Fringe Feature Extraction and Matching

In this section, we need to achieve precise matching between actual interferometric fringes and reference interferometric fringes. In interference fringe matching, due to the influence of drift errors, flight attitude errors, and interference parameter errors, the interference fringe will produce geometric distortion and nonlinear phase errors, so the algorithm based on template matching is not applicable under such conditions. On the other hand, deep learning-based methods are not suitable since there is no relevant interference fringe database. Therefore, we adopt a feature-based image matching algorithm framework.

The classical SIFT algorithm [19] establishes a linear scale space using Gaussian filters. However, it may not preserve significant phase changes and edge information in the interferometric fringes, affecting the detection of feature points. Instead, we use non-linear diffusion filtering to construct the scale space. When applied to images, non-linear diffusion filtering [20] represents the brightness of the image through the divergence of the flow function. Non-linear diffusion filtering is expressed by a partial differential equation as follows:

$$\varphi_t = \frac{\partial \varphi}{\partial t} = div(c(x, y, t) \cdot \nabla \varphi) \tag{7}$$

Here, $\varphi$ represents the interferometric phase value, $t$ is the diffusion time, $div$ and $\nabla$ are the divergence and gradient operators, and $c(x, y, t)$ is the conductivity function. By choosing an appropriate conductivity function, diffusion can be adapted to the local structure of the image. Perona and Malik proposed a way to construct the conductivity function [21]:

$$c(x, y, t) = g(|\nabla \varphi_\sigma(x, y, t)|) \tag{8}$$

Here, $\nabla \varphi_\sigma$ is the gradient of the interferometric phase $\nabla \varphi_\sigma$ after Gaussian smoothing, and the function $g$ is defined as:

$$g(\nabla \varphi) = \frac{1}{1 + (|\nabla \varphi|/K)^2} \tag{9}$$

Here, $K$ is the contrast factor, controlling the diffusion level. A larger value retains less edge information. By solving the non-linear partial differential equation using the Additive

Operator Splitting (AOS) algorithm, we obtain images corresponding to different times in the non-linear scale space:

$$\varphi^{i+1} = \left( I - (t_{i+1} - t_i) \cdot \sum_{i=1}^{m} A_l\left(\varphi^i\right) \right)^{-1} \varphi^i \tag{10}$$

Here, $I$ is the identity matrix, and $A_l$ represents the derivative along the $I$ direction. After establishing the nonlinear scale space, all pixel points were compared with the surrounding and adjacent pixel points in the two layers above and below. After obtaining feature points, a three-dimensional quadratic function was used for fitting to obtain subpixel positioning results for feature points. The use of nonlinear diffusion filtering to extract feature points in the nonlinear scale space can suppress noise while preserving edge information, thereby increasing the quantity and accuracy of feature points. As shown in Figure 6, a comparison was made between feature points detected in the linear scale space using the SIFT algorithm and those detected using nonlinear diffusion filtering. It can be observed that the SIFT algorithm detects fewer feature points, and edge points and corners on the fringe map are not sufficiently detected under the smoothing effect of Gaussian linear space. In contrast, the feature points detected using the nonlinear diffusion filter retain edge points and corners, resulting in a higher number of detected feature points.

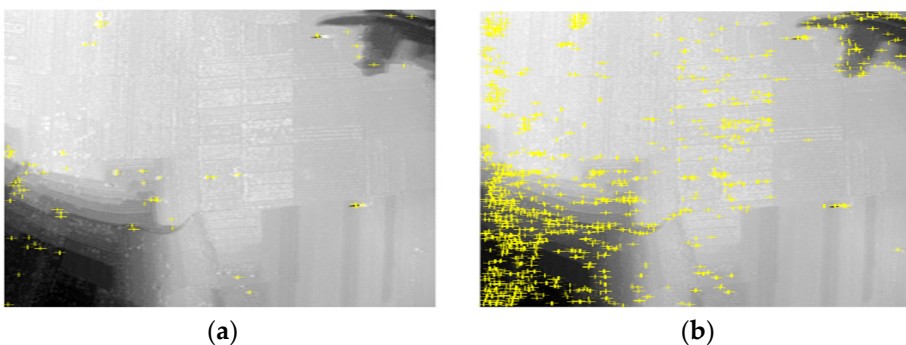

|   (a)   |   (b)   |

**Figure 6.** Comparison of feature point detection methods, where feature points are represented in yellow. (**a**) Detecting feature points using the SIFT algorithm. (**b**) Detecting feature points in nonlinear scale space.

After determining the positions of feature points, a selection process is applied using the interferometric coherence coefficient to filter out feature points with low coherence. A descriptor is then constructed to form a subset of features. Once key points are selected in the scale space, these key points, being extrema in the scale space, undergo wavelet transformation [22–25] to separate multiscale features around the key points, achieving effective feature extraction. Rectangular sub-images are established with key points as centers to describe key point information using sub-region image features, enabling mutual matching between key points in two images. Two layers of 2D discrete wavelet transforms are performed on the sub-region around key points. Since interference fringes contain a significant amount of noise in high-frequency components and texture information in low-frequency components, the first layer's low-frequency components and the second layer's low-frequency components are used for feature extraction. Simultaneously, low-frequency components obtained through wavelet decomposition possess translation invariance, enhancing the robustness of fringe matching and making it more resistant to noise, distortions, and other interferences.

After removing the flat ground effect from the interference fringes, which reflects terrain undulation information with a gradually changing gradient, analyzing local gradient information around feature points can effectively describe the current feature points. Histogram of oriented gradient (HOG) is a technique used for texture-based image analysis [24–26], simplifying images by extracting gradient information. HOG extracts features

that have a locally distinctive shape based on edges or gradient structures. Through block operations, HOG makes local geometric and intensity changes easy to control. If the translation or rotation of an image is much smaller than the scale of local spatial and directional units, the detected differences are minimal. Detecting HOG features in the wavelet domain of two heterogeneous images can effectively describe the texture information around feature points in both images, making it suitable for matching between two images with rich texture features.

First, the low-frequency approximate image extracted through wavelet transformation is divided into blocks. The gradients $G_x$ $G_y$ in the $x$ and $y$ directions for each pixel in each block are calculated using gradient templates:

$$\begin{cases} H_x = \begin{bmatrix} -1 & 0 & 1 \end{bmatrix} \\ H_y = \begin{bmatrix} -1 & 0 & 1 \end{bmatrix}^T \end{cases} \tag{11}$$

For the input image $I$, performed convolution operations with the template, respectively, to obtain the gradients $I_x$ and $I_y$ in the $x$ and $y$ directions:

$$\begin{cases} I_x = I * H_x \\ I_y = I * H_y \end{cases} \tag{12}$$

Thus, we obtained the magnitude $|G| = \sqrt{I_x{}^2 + I_y{}^2}$ and orientation $\theta = \arctan(I_y/I_x)$ of the gradient. Then, we divided 0–360° into several ranges, counted the occurrences of gradient directions in different ranges, and formed a histogram of gradient directions. We connected the histograms of gradient directions for each block region to create the HOG feature. This process resulted in a feature vector for each keypoint's surrounding region in the wavelet domain. The first-level low-frequency components were divided into 4 × 4 cells, and the orientation gradients were calculated for each block region to form a histogram of gradient directions. The second-level low-frequency components were divided into 8 × 8 cells, aiming to increase gradient information around key points and improve matching accuracy. Figure 7 clearly shows the process of descriptor construction.

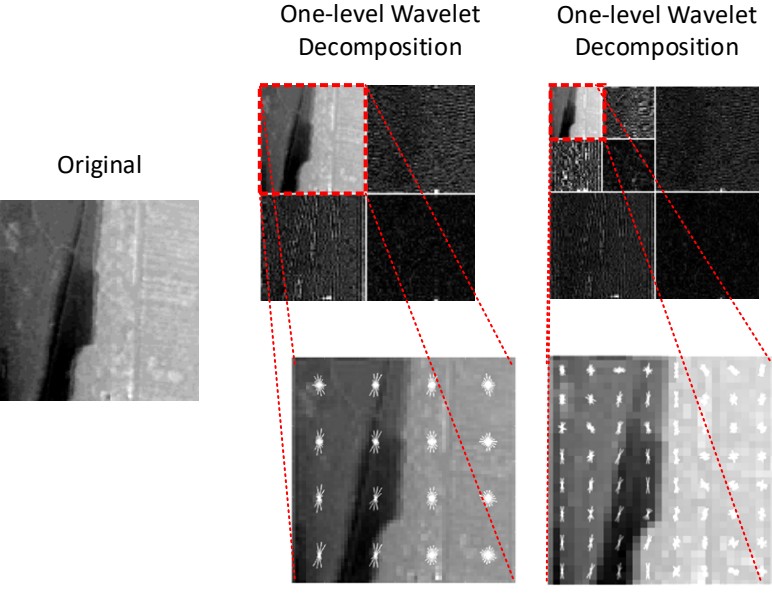

**Figure 7.** Illustration of feature descriptor construction.

Once the descriptor was constructed, the next step involved using the minimum Euclidean distance method for real-time matching of feature vectors between actual and

reference interferograms. This process eliminates outlier matches, resulting in pairs of matching points. The three-dimensional position information of the matching points can be obtained by querying latitude, longitude, and height information. The algorithmic flowchart for interferometric fringe feature matching is illustrated in Figure 8, with the technical process on the left and key step results on the right.

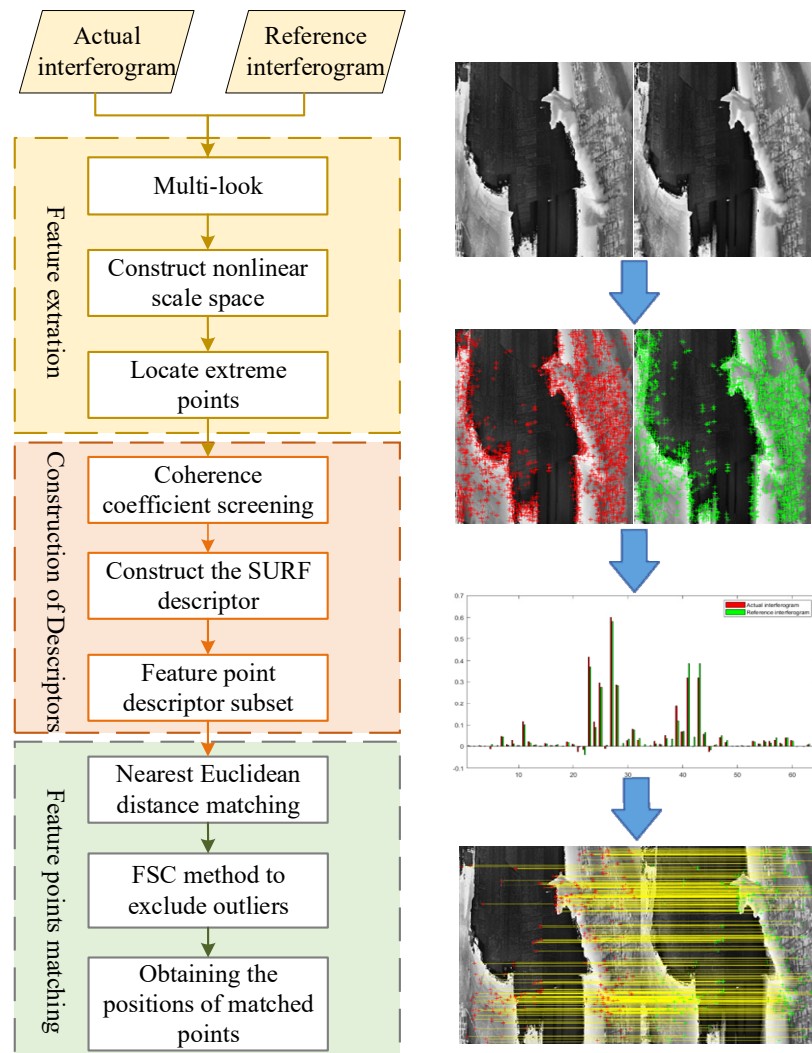

**Figure 8.** InSAR interferogram matching algorithm flow.

## 5. Platform Three-Dimensional Positioning Model Based on InSAR Geometry

Through the matching of the interferograms, the corresponding position and elevation information for the tie points was obtained, and we established the inversion model of the platform position as shown in Figure 9. The *x*-axis represents the flight direction of the platform, the *z*-axis represents the elevation direction, and the *y*-axis forms a right-handed coordinate system with the *x*-axis and *z*-axis. $A_1$ and $A_2$ represent the positions of the master and slave radar antennas, respectively, *P* represents the matched tie point, and **R** is the slant range vector from the master antenna to point *P*, referred to as the look vector. **b** is the baseline vector, where $b_n$ represents the length of the cross-track baseline and $b_v$ represents the length of the along-track baseline. Orthogonal unit vectors constitute a moving coordinate system, where $\hat{v}$ is in the flight direction, $\hat{n}$ is in the same direction as the cross-track baseline, and $\hat{w}$ is determined using the right-hand rule.

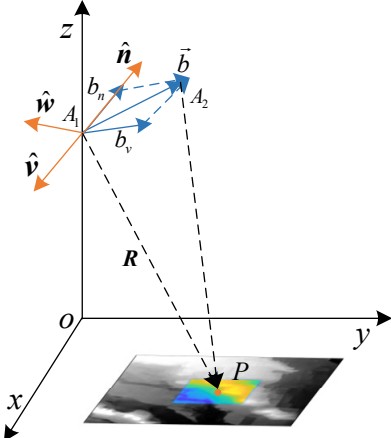

**Figure 9.** Three-dimensional position inversion model of the platform.

In the moving coordinate system, the unit look vector can be represented as:

$$\hat{r} = \mu\hat{v} + \eta\hat{n} + \varsigma\hat{w} \tag{13}$$

where $\mu = <\hat{r},\hat{v}>$, $\eta = <\hat{r},\hat{n}>$, $\varsigma = <\hat{r},\hat{w}>$, and $< \cdot >$ represents the inner product operation between vectors. Therefore, the representation of the unit look vector in the moving coordinate system can be calculated as [27–29]:

$$\hat{r}_{vnw} = \begin{bmatrix} \mu \\ \eta \\ \varsigma \end{bmatrix} = \begin{bmatrix} \frac{\lambda f_{dc}}{2v} \\ -\frac{\lambda \varphi_a}{2\pi Q b_n} - \frac{b_v}{b_n}\frac{\lambda f_{dc}}{2v} \\ \pm\sqrt{1 - \mu^2 - \eta^2} \end{bmatrix} \tag{14}$$

where $f_{dc}$ is the Doppler centre frequency, $v$ is the platform velocity. The positive or negative sign is determined by the side-looking direction of the radar. The unit look vector is transformed from the moving coordinate system to the carrier coordinate system using the transformation matrix $T$:

$$\hat{r} = \mathbf{T} \times \hat{r}_{vnw} \tag{15}$$

where transformation matrix **T** is given by:

$$\mathbf{T} = \begin{bmatrix} 1 & 0 & 0 \\ 0 & \cos\theta_b & -\sin\theta_b \\ 0 & \sin\theta_b & \cos\theta_b \end{bmatrix} \tag{16}$$

where $\theta_b$ is the angle between the baseline and the *x-y* plane. Therefore, the three-dimensional position vector of the platform *A* can be represented as:

$$\begin{aligned} A &= P - R \times \hat{r} \\ &= P - R \times \begin{bmatrix} \frac{\lambda f_{dc}}{2v} \\ \left(-\frac{\lambda\varphi}{2\pi Q b_n} - \frac{b_v}{b_n}\frac{\lambda f_{dc}}{2v}\right)\cos\theta_b \mp \sqrt{1-\mu^2-\eta^2}\sin\theta_b \\ \left(-\frac{\lambda\varphi}{2\pi Q b_n} - \frac{b_v}{b_n}\frac{\lambda f_{dc}}{2v}\right)\sin\theta_b \pm \sqrt{1-\mu^2-\eta^2}\cos\theta_b \end{bmatrix} \end{aligned} \tag{17}$$

## 6. Experiments and Results

We validated the proposed method through semi-physical experiments using data acquired by a dual-antenna Ku-band FMCW InSAR system developed by the Institute of Electronics, Chinese Academy of Sciences. The InSAR system was mounted on an unmanned aerial vehicle (UAV). The InSAR system is shown in Figure 10, and Table 1 provides the airborne InSAR flight parameters.

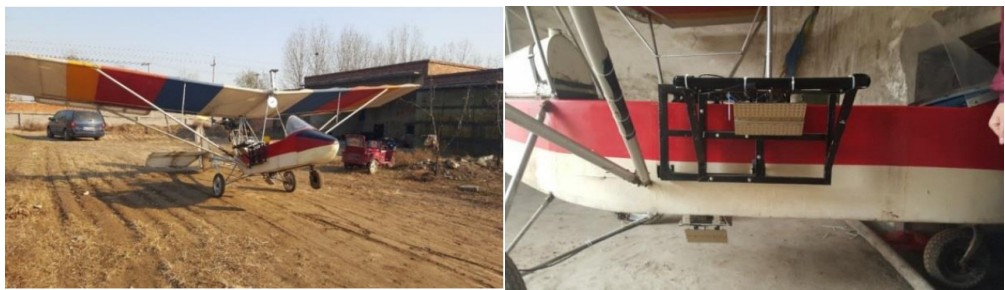

**Figure 10.** Ku-FMCW InSAR system.

**Table 1.** Actual airborne flight parameters.

| Parameter | Value |
|---|---|
| Carrier frequency | 14.5 GHz |
| Platform height | 1898.86 m |
| Nearest slant range | 1672.661 m |
| Baseline length | 0.4337 m |
| Baseline inclination angle | 41.9627° |
| Squint angle | 5.2° |

The radar system is equipped with two position and orientation systems (POS): POS610 and Mini-POS. Mini-POS is a lightweight and compact system with lower accuracy. We conducted matching and positioning experiments using the low-precision IMU from Mini-POS, which has an accelerometer drift error of 0.1 mg and a gyroscope drift error of 1°/h. Our method was validated using high-precision positioning data from POS610. To improve the accuracy of matching and positioning, we utilized light detection and ranging (LiDAR) DEM data with a grid resolution of 1 m as a reference DEM.

### 6.1. Generation of Reference Interferograms

Figure 11 depicts the imaging area determined by the reference trajectory of the IMU for the reference DEM. The imaging area is highlighted within the red box. Figure 12a shows the range image after range resampling, Figure 12b displays the computed phase map, Figure 12c illustrates the interferogram with the removed ground phase, and Figure 12d presents the reference interferogram with added wrapped phase for matching.

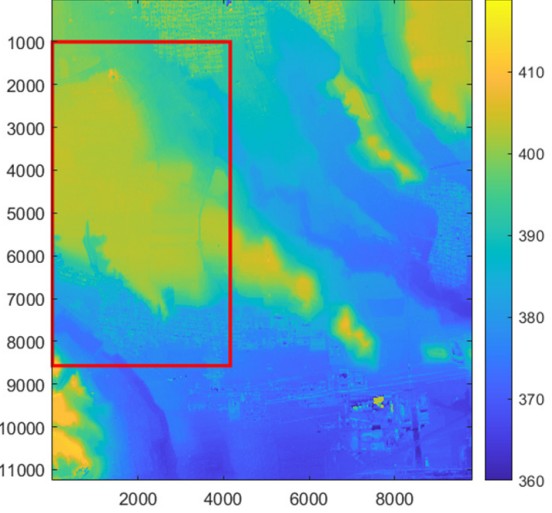

**Figure 11.** The reference DEM determines the imaging area based on the IMU reference trajectory. The imaging area is marked within the red box area.

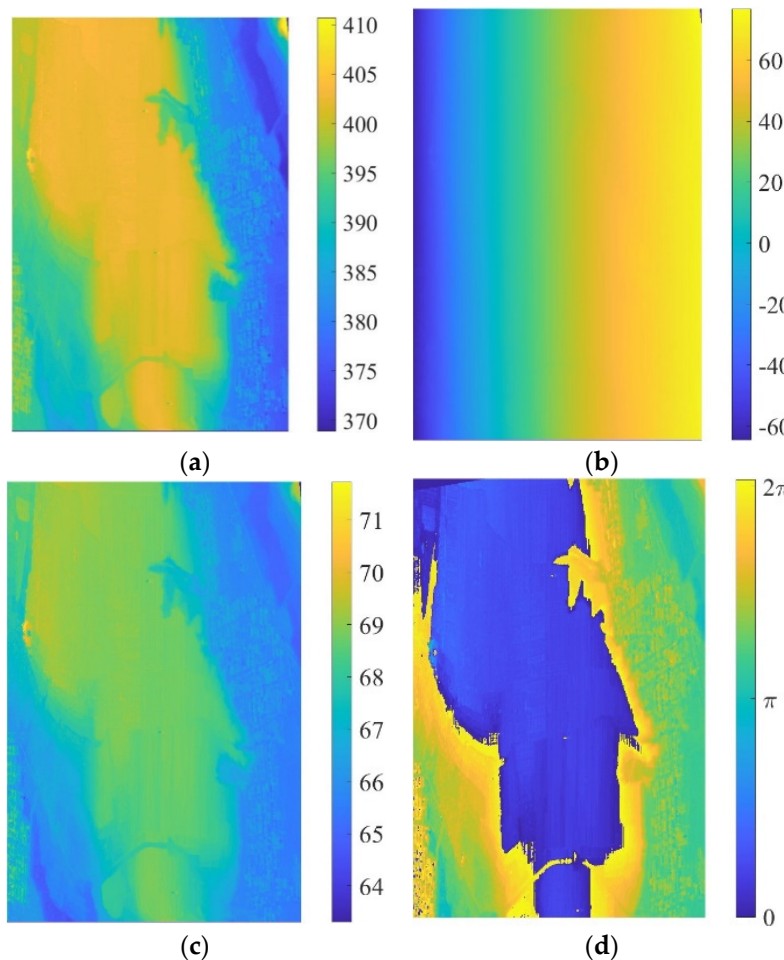

**Figure 12.** Processing steps for the reference interferogram. (**a**) The reference DEM undergoes slant range resampling. (**b**) Interferometric phase map. (**c**) Interferometric phase map after removing the flat Earth phase. (**d**) Interferogram with wrapped phase.

### 6.2. Generation of Actual Interferograms

Real-time generation of InSAR interferograms mainly involves dual-antenna SAR imaging, self-focusing, registration, interferometric processing, and ground removal. These processing techniques are well-established, and we present the interferogram processing steps and results for our data directly. Figure 13a shows the interferogram before registration, Figure 13b displays the interferogram after fine registration, Figure 13c illustrates the interferogram after filtering, and Figure 13d presents the interferogram after removing the flat Earth phase.

### 6.3. Interferogram Matching and Platform Localization Results

During the flight, due to interruptions in GNSS signals, there was no real-time correction applied to the IMU, leading to drift errors. Therefore, the purely inertial navigation solution using Mini-POS causes the position error to increase over time. As a result, using its localization information for navigation introduces position offsets, causing misalignment between actual interferograms and reference interferograms. The results obtained after applying our proposed interferogram feature matching algorithm are shown in Figure 14, where the left side depicts actual generated interferograms, and the right side shows reference interferograms. And, after using the ransac algorithm to remove outliers, the correct matching point pair was obtained.

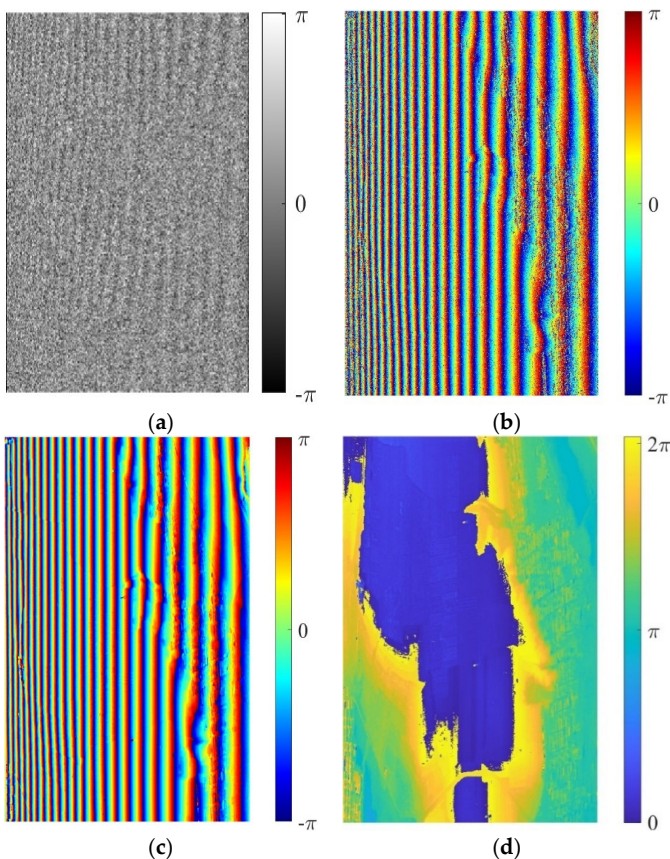

**Figure 13.** Actual interferogram generation process. (**a**) Interferogram before registration. (**b**) Interferogram after registration. (**c**) Interferogram after filtering. (**d**) Interferogram after removing the flat Earth phase.

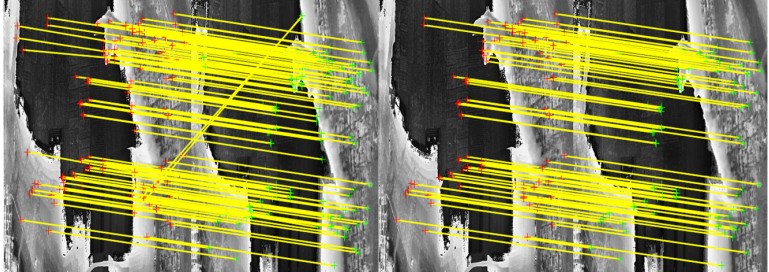

**Figure 14.** Interferogram matching results (**left**: actual generated interferogram, **right**: reference interferogram). Interference fringe matching results with external points removed.

The reference interferogram provides queryable location and elevation information for each matching point, allowing us to obtain three-dimensional position information for the matching points. Utilizing the three-dimensional localization model proposed in Section 5, we calculated the three-dimensional position of the platform. Figure 15 illustrates the differences between the positions obtained by the InSAR matching positioning system and the high-precision POS610 system at each matching point. It can be observed that the *x*-axis error is within 3 m, the *y*-axis error is within 7 m, and the *z*-axis error is within 2 m, resulting in a total positioning error within 8 m. The maximum error in the *y*-axis is attributed to the imprecise roll angle of the platform. In subsequent experiments, we plan to calibrate the interferometric parameters using the three-dimensional information of the matching points to achieve higher positioning accuracy. The minimum positioning error is observed in the *z*-axis, highlighting the advantages of interferometric technology in elevation measurement.

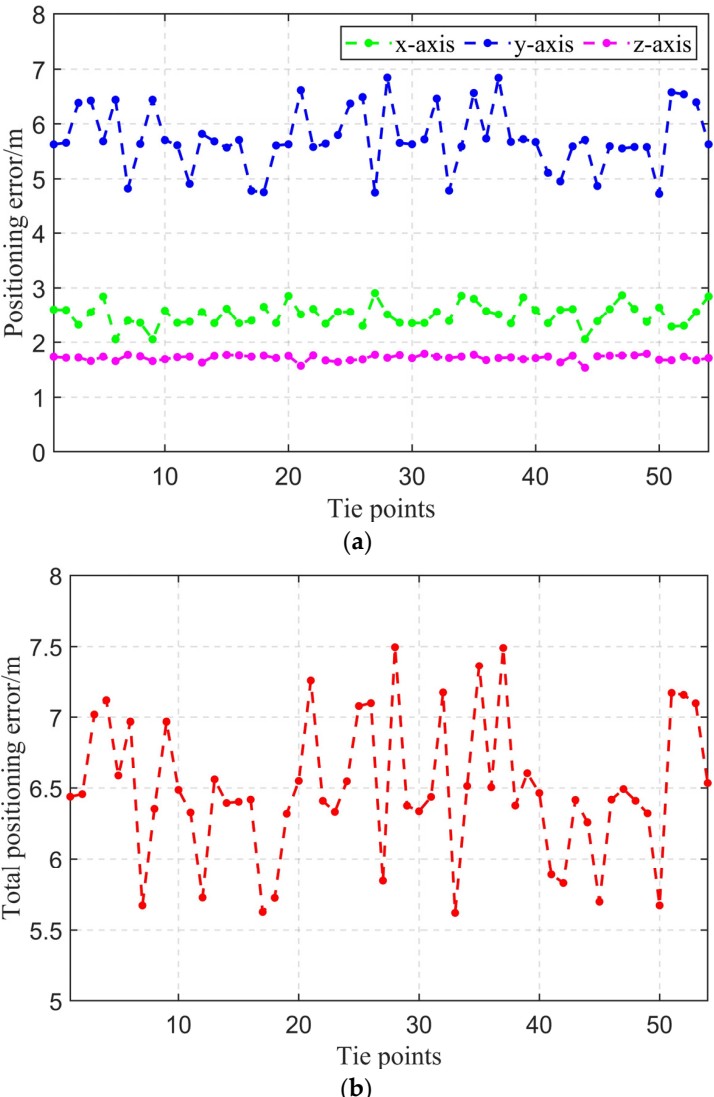

**Figure 15.** Positioning error of each axis and total positioning error. (**a**) Positioning error for each axis, (**b**) Total positioning error.

To conduct a more in-depth analysis of the data, we employed statistical indicators such as maximum error, average error, and RMSE to evaluate positioning accuracy. As shown in Table 2, the three-dimensional positioning using InSAR interferograms has a maximum error of 7.49 m, an average error of 6.50 m, and an RMSE of 6.51 m. The positioning accuracy restricted to being within 10 m demonstrates that the algorithm achieved high precision in matching navigation. It is worth noting that our experiment is a semi-physical simulation, utilizing a high-precision LiDAR DEM as a reference DEM. In the next steps, we plan to implement a specific hardware system for corresponding positioning experiments.

**Table 2.** Positioning error statistical results of different axes.

| Direction | Maximum Error [m] | Average Error [m] | RMSE [m] |
|:---:|:---:|:---:|:---:|
| $x$ | 2.91 | 2.50 | 2.51 |
| $y$ | 6.89 | 5.73 | 5.76 |
| $z$ | 1.79 | 1.72 | 1.72 |
| total | 7.49 | 6.50 | 6.51 |

## 7. Discussion

In this paper, we introduced the application of interferometry techniques to the method of positioning for airborne SAR platforms, utilizing interferograms for matching and positioning. This approach combines the advantages of traditional SAR images and terrain matching positioning methods, leveraging the long-term elevation stability provided by DEMs generated through InSAR mapping, as well as the three-dimensional positioning capabilities of InSAR, thereby improving the final positioning accuracy to be within 10 m. The high precision of InSAR fringe matching is primarily due to the additional information in the elevation dimension provided by InSAR compared to SAR, enhancing the positioning capability from two dimensions to three dimensions. On the other hand, InSAR fringes reflect the undulating nature of the terrain, offering more stability compared to SAR images and thus providing higher stability in the matching process. The error of the results is explained below and some issues are discussed.

### 7.1. Positioning Error Analysis

The errors in InSAR interference fringe matching navigation mainly include matching errors, platform attitude angle errors, and interference system parameter errors. The interference parameter errors can be eliminated after calibration, so we will not go into details here. Therefore, this section discusses the matching error and platform attitude angle error and their impact on platform positioning, and conducts an error analysis based on the actual parameters of airborne InSAR.

#### 7.1.1. Platform Attitude Angle Error

Figure 16 illustrates the positioning error in a plane under the presence of yaw angle error. Since the attitude angle mainly affects the accuracy of plane positioning, an analysis of positional offsets in the azimuthal and range directions is conducted. In the diagram, the aircraft flies along the ideal trajectory. The actual trajectory and the beam illumination area when there is a certain yaw angle error, denoted as $\theta_y$, the actual are indicated in red. The platform flies along the azimuthal direction $x$-axis, range direction $y$-axis, and vertical direction $z$-axis, with $H$ representing the platform's flying height. Neglecting terrain fluctuations, the actual slant range matches the ideal trajectory's slant range, denoted as $R_0$. The initial look angle is $\theta$, the actual look angle is $\theta_1$, and the slant angle is $\theta_{sq}$. The positioning offset is represented as $X_g$.

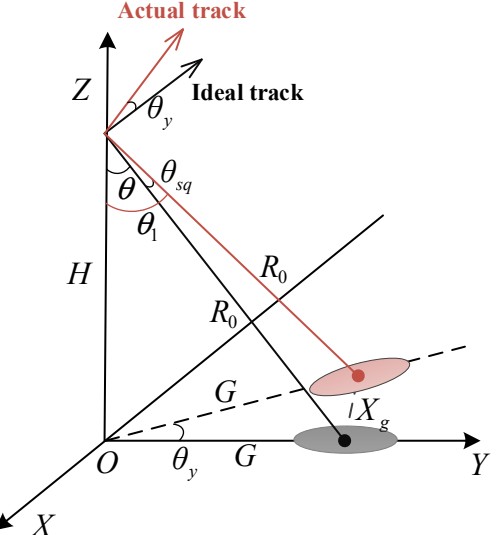

**Figure 16.** Effect of yaw angle error on positioning.

The slant range $R_0$, initial range positioning distance $G$, $X_g$, and $\theta_{sq}$ are derived from the geometric relationships in the diagram.

$$\begin{cases} R_0 = \dfrac{H}{\cos\theta} \\ G = H\tan\theta \\ X_g = G\sqrt{2 - 2\cos\theta_y} \\ \theta_{sq} = \arccos\left(1 - \dfrac{X_g^2}{2R_0^2}\right) \end{cases} \tag{18}$$

Therefore, the positioning errors in the azimuthal direction $\Delta x$ and in the range direction $\Delta y$ are:

$$\begin{cases} \Delta x = G\sin\theta_y = H\tan\theta\sin\theta_y \\ \Delta y = H\tan\theta\left(\cos\theta_y - 1\right) \end{cases} \tag{19}$$

From the equation, we can determine the influence of yaw angle error on positioning offsets. Using the airborne flight parameters provided in this paper for simulation, with a platform flying height of 1898.86 m and a yaw angle range set to $[-0.2, 0.2]$ degrees, and considering the actual downward viewing angle, which ranges from $30°$ to $60°$ from near slant range to far slant range. The results are shown in Figure 17. It can be observed that the azimuthal positioning error is greatly influenced by yaw angle error, and this influence becomes more pronounced as the downward viewing angle increases. However, the impact on range direction positioning is minimal.

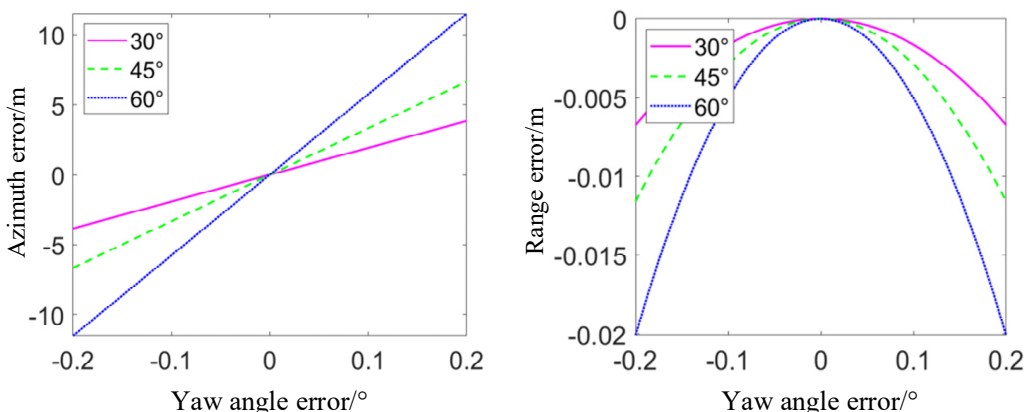

**Figure 17.** Yaw angle error and positioning offset relationship.

Figure 18 depicts the conceptual illustration of positioning errors in a plane under the presence of pitch angle error. Similar to the analysis method for yaw angle, the aircraft flies along the ideal trajectory. The actual trajectory and the beam illumination area when there is a certain pitch angle error, denoted as $\theta_p$, are depicted in red. The platform flies along the azimuthal direction $x$-axis, range direction $y$-axis, and vertical direction $z$-axis, with H representing the platform's flying height. Initially, under the ideal conditions, the slant range is denoted as $R_0$, while after introducing yaw angle error, it is represented as $R$. The initial downward viewing angle is $\theta$, the actual downward viewing angle is $\theta_1$, and the slant viewing angle is $\theta_{sq}$. The positioning offset is represented as $X_g$.

The slant range $R_0$, initial range positioning distance $G$, and $X_g$ are derived from the geometric relationships depicted in the diagram.

$$\begin{cases} R_0 = \dfrac{H}{\cos\theta} \\ G = arc\cos\left(\dfrac{H}{R\cos\theta_p}\right) \\ X_g = H\tan\theta_p = R\sin\theta_{sq} = \dfrac{H}{\cos\theta}\tan\theta_{sq} \end{cases} \tag{20}$$

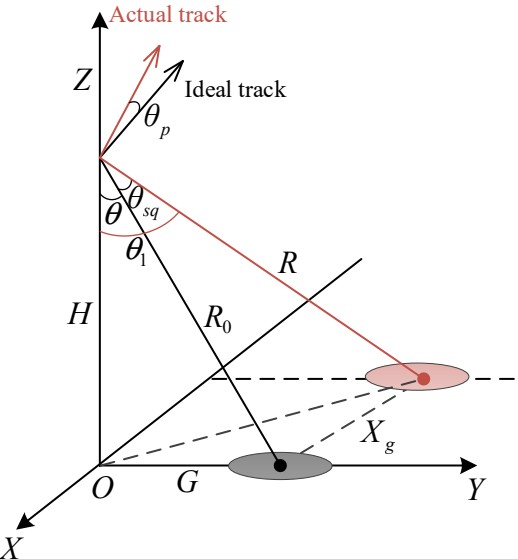

**Figure 18.** Effect of pitch angle error on positioning.

Therefore, the positioning errors in the azimuthal direction, represented by Δx, and in the range direction, represented by Δy, are:

$$\begin{cases} \Delta x = X_g = H \tan \theta_p \\ \Delta y = \frac{H}{\cos \theta_p} \tan \theta - \tan \theta \end{cases} \tag{21}$$

From the equation we can determine the influence of pitch angle error on positioning offsets. Using the airborne flight parameters provided in this paper for simulation, with a platform flying height of 1898.86 m and a yaw angle range set to [−0.2, 0.2] degrees, and considering the actual downward viewing angle, which ranges from 30° to 60° from the near slant range to the far slant range. The results are shown in Figure 19. It can be observed that pitch angle mainly affects azimuthal positioning error and remains relatively constant regardless of changes in the downward viewing angle. The impact on range positioning is minimal.

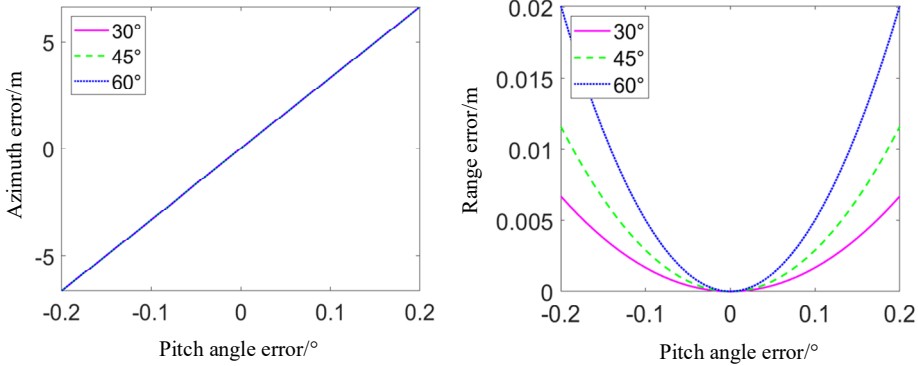

**Figure 19.** The influence of pitch angle on azimuth and range positioning.

### 7.1.2. Fringe Matching Error

Matching errors mainly stem from two aspects: the accuracy of the reference map and the accuracy of the matching algorithm. The spatial resolution and elevation accuracy of the reference map are critical factors determining its quality. A lower spatial resolution implies that each pixel in the image represents a larger actual ground area, which may result in the inability to accurately identify smaller ground features during the matching process, thereby increasing positioning errors. Additionally, if the reference map exhibits

geometric distortions, such as shape distortions caused by imaging angles or differences in ground heights, it directly affects the accuracy of positioning. These geometric distortions may cause the matching algorithm to fail to correctly align the reference map with the actual observed image, introducing additional positioning errors. Moreover, inaccuracies in feature point or region extraction during the matching algorithm's process can also lead to misalignment and matching errors.

According to the geometric positioning model, the matching error transfer relationship in the $x$ direction, $y$ direction, and $z$ direction is:

$$\begin{cases} \Delta P_x = \Delta A_x \\ \Delta P_y = \Delta A_y \\ \Delta P_z = \Delta A_z \end{cases} \tag{22}$$

That is, the matching error and the reference DEM error directly affect the positioning error. Therefore, improving the accuracy of the matching algorithm and using high-precision reference maps can increase the positioning accuracy in equal proportions.

### 7.2. High-Precision Needs and Future Prospects

The generation of interference fringes is a basic and critical step. It is worth mentioning that in the actual navigation process, the system is a continuous iterative process. After the satellite signal is rejected, the InSAR matching navigation process is started in time, and combined filtering with the INS is performed to obtain the corrected position and attitude information. And imaging, matching, and positioning are continuously performed to reasonably control the interval, accurate position, and attitude information. In addition, even in the absence of precise positioning information to achieve high-precision imaging, image generation and interference processes can be achieved through self-focusing algorithms to obtain interference fringes.

In InSAR matching positioning, the interferometric baseline parameters have a significant impact on the generation of interferograms and the accuracy of three-dimensional positioning, leading to considerable nonlinear distortion in the phase of the reference interferogram and significant errors in the three-dimensional positioning. Therefore, calibration of the interferometric parameters is necessary to ensure high positioning accuracy. Our method involves using matching points on the reference DEM as control points and optimizing the interferometric parameters using the least squares method to ensure the accuracy of these parameters.

Our next step is to optimize the efficiency of the program and adopt parallel processing to meet the needs of real-time operation on airborne platforms, aiming to implement a complete positioning and navigation system.

### 8. Conclusions

In this article, we propose a novel platform positioning and navigation method. The InSAR interferogram matching positioning and navigation system combines the advantages of traditional SAR images and terrain matching positioning methods, and achieves higher positioning accuracy. In our paper, we have provided detailed steps, including the generation of reference interferograms, extraction and matching of interferogram features, and the establishment of a three-dimensional positioning model for the platform. Finally, the effectiveness of our proposed method was verified through actual flight experiments. Future research will focus on improving the algorithm's operational efficiency to meet the real-time requirements of airborne systems. On the other hand, it will be necessary to enhance the accuracy of interferometric parameters through interferometric calibration techniques, thereby achieving higher precision in positioning capabilities.

**Author Contributions:** Conceptualization, L.L. and B.W.; methodology, B.W.; software, L.L.; validation, L.L., Y.W. and B.W.; formal analysis, M.X.; investigation, L.L.; resources, B.W.; data curation, L.L.; writing—original draft preparation, L.L.; writing—review and editing, L.L. and B.W.; visualization,

L.L.; supervision, B.W.; project administration, M.X.; funding acquisition, B.W. All authors have read and agreed to the published version of the manuscript.

**Funding:** This work was supported by the National Natural Science Foundation of China under Grant 62073306.

**Data Availability Statement:** The data presented in this study are available on request from the corresponding author. The data are not publicly available due to confidentiality of the data.

**Acknowledgments:** The authors thank the staff of the National Key Laboratory of Microwave Imaging Technology, Aerospace Information Research Institute, Chinese Academy of Sciences, for their valuable conversations and comments.

**Conflicts of Interest:** The authors declare no conflicts of interest.

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
