# Peer review of "Airborne Platform Three-Dimensional Positioning Method Based on Interferometric Synthetic Aperture Radar Interferogram Matching"

_remotesensing, doi:10.3390/rs16091536_

Round 1
Reviewer 1 Report
Comments and Suggestions for Authors
General comments:
The paper proposes an airborne platform positioning method based on interferometric SAR (InSAR) interferogram matching, in which the real-time InSAR-processed interferograms are matched with the simulated interferograms using a digital elevation model (DEM), and the 3D position information of the matched points is obtained. Subsequently, the 3D positioning model of the platform is established using the unit line-of-sight vector decomposition method. In short, a new idea is provided for airborne platform positioning.
Specific comments:
1. The positioning of platform is only within 10 meter, can you discuss the factors affecting the positioning accuracy quantitatively in terms of DEM precision, matching accuracy etc.
2. How can you get fine interferogram fringe without high precision positioning of SAR platform? Please discuss it or give more in-depth information on this.
3. The elevation range of the experimental area is only from 370 m to 410 m, which is relatively small. Why don’t you choose one large elevation difference region for the test? In my view, the larger elevation difference, the high precision of platform estimation.
4. On other hand, in the mountainous areas with undulating terrain, the difference in elevation is large, and the interferometric fringes are more complicated and the geometric distortion is more serious, so please discuss it when it is used in the mountainous areas.
5. How can you verify your results with simulation or with a priori information?
Comments on the Quality of English LanguageModerate editing of English language required.
Reviewer 2 Report
Comments and Suggestions for Authors
Some sentences should be checked and improved.
Round 2
Reviewer 1 Report
Comments and Suggestions for Authors
My questions have been replied well, so I have no any more questions.